# Hepatitis B vaccination coverage, knowledge and sociodemographic determinants of uptake in high risk public safety workers in Kaduna State, Nigeria: a cross sectional survey

Chinwe Lucia Ochu,[1] Caryl M Beynon[2]

► Prepublication history and additional material are available. To view these files please visit the journal online (http://dx.doi.org/10.1136/bmjopen-2017-015845).

## ABSTRACT

**Objectives** To estimate hepatitis B vaccination (HBVc) coverage, and knowledge and sociodemographic determinants of full dose uptake in Federal Road Safety Corps (FRSC) members, Kaduna State, Nigeria, to inform relevant targeted vaccination policies.

**Design** A cross sectional survey of FRSC members, Kaduna Sector Command.

**Settings** Six randomly selected unit commands under Kaduna Sector Command, Kaduna State, Nigeria.

**Participants** A pilot tested, structured, self-administered questionnaire was administered to 341 participants aged ≥18 years with ≥6 months of service between 17 June and 22 July 2015. Excluded were FRSC members in road safety 1 zonal command headquarters as the zonal command includes other states beyond the study scope.

**Primary outcome** HBVc status of participants categorised as 'not vaccinated' for uptake of <3 doses and 'vaccinated' for uptake of ≥3 doses.

**Analysis** Descriptive analysis estimated HBVc coverage while logistic regression ascertained associations.

**Results** Most participants were men, aged 30–39 years, with 3–10 years of service and of marshal cadre. HBVc coverage was 60.9% for ≥1 dose and 30.5% for ≥3 doses. Less than 47% of participants scored above the mean knowledge score for hepatitis B virus (HBV) and HBVc. Female sex (AOR 2.28, 95% CI 1.15 to 4.52, p<0.05), perceiving there to be an occupational risk of exposure to HBV (AOR 2.86, 95% CI 1.06 to 7.70, p<0.001) and increasing HBVc knowledge (AOR 2.68, 95% CI 1.83 to 3.92, p<0.001) were independent predictors of full dose HBVc in FRSC members, Kaduna Sector Command.

**Conclusions** HBVc coverage and knowledge were poor among FRSC members, Kaduna Sector Command. Educational intervention, geared towards improving FRSC members' knowledge of HBVc and perception of risk of occupational exposure to HBV, is recommended for these vulnerable public safety workers. Such enlightenment could be a cheap and easy way of improving HBVc coverage in the study population.

[1]Family Medicine, Ahmadu Bello University Teaching Hospital, Zaria, Nigeria
[2]Public Health, University of Liverpool, Liverpool, UK

**Correspondence to**
Dr Chinwe Lucia Ochu;
lucia.ochu@online.liverpool.ac.uk

### Strengths and limitations of this study

► This is the first study to estimate hepatitis B vaccination coverage of public safety workers such as the Federal Road Safety Corps in Nigeria, despite these workers being occupationally exposed to hepatitis B virus.

► The participating unit commands were randomly selected and the study had a high response rate, hence minimising selection bias and improving the generalisability of the research findings.

► Retrospective studies are prone to recall bias; this was mitigated in this study by omitting inconsistent data suggestive of guessing at the analysis stage.

► Researcher bias was mitigated by the use of a pre-validated data collection instrument and by predetermining analytical strategies before data collection while confounding was minimised through multivariate analysis.

► Missing data made sample size in multivariate analysis less than the pre-research estimate, although the proportion analysed constituted a good representation of the entire study population.

## INTRODUCTION

Hepatitis B virus (HBV) is a highly infectious blood borne pathogen, usually transmitted via percutaneous or mucosal exposure to infected blood and body fluids.[1] HBV infection affects about one-third of the world's population, with >350 million persons being chronic carriers.[2] [3]Sub-Saharan Africa and Southeast Asia have the highest prevalence of chronic HBV (about 10–20%).[2] HBV infection has heterogeneous outcomes: acute viral hepatitis, spontaneous clearance or chronicity, with its common fatal sequelae of hepatic cirrhosis and hepatocellular carcinoma (HCC).[3] [4] Most adult onset infections resolve spontaneously, with only 5–10% resulting in chronic carriership.[2] Chronicity is commonly associated with early childhood exposures, with an estimated 90% of perinatal transmissions becoming chronic infections.[4] Perinatal and horizontal transmissions are

the predominant routes of HBV infection in hyperendemic settings.[5]

Hepatitis B vaccination (HBVc) is the most effective way of controlling HBV infection.[6] HBV control in sub-Saharan Africa targets mother to child transmission via HBVc of children aged 0–5 years.[2 7] Although chronicity has been the major HBV outcome of public health interest, recent subtle transitions in the global mortality burden of HBV outcomes demand readjustment of this focus. In a comparative systematic analysis of global disease burden, Lozano *et al* demonstrated the trend in HBV related outcomes between 1990 and 2010.[8] Although HBV related HCC caused more deaths than acute HBV infection, the percentage increase in age standardised death rates was about 11 times higher for acute HBV infection (29.2%) than for HBV related HCC (2.6%) while death from HBV related liver cirrhosis declined by 18.5%.[8] This growing mortality trend for acute HBV infection demands a renewed public health action in addressing this often neglected outcome of HBV.

Prevention strategies for HBV should also target those at high risk of acute infections. Public safety workers (PSWs), such as fire fighters, correctional officers, rescue workers and emergency medical service providers with regular exposure to blood or body fluids, have similar risks as hospital based healthcare workers (HCWs) of contracting HBV.[9] Also, HCWs or PSWs, in the course of their duties, can infect children who consequently become chronic carriers. Controlling HBV infection in HCWs and PSWs is therefore of public health relevance. The WHO prescribes universal HBVc of HCWs and PSWs with frequent blood–skin exposure.[10] A standard three dose vaccine regimen, with the second and third doses given 1 month and 6 months apart from the initial dose respectively, is very effective in conferring immunity against HBV for ≥20 years.[1 11]

Nigeria is hyperendemic for HBV; Schweitzer *et al* reported a pooled HBV prevalence estimate of 9.76% (95% CI 9.59 to 9.93).[5] This hyperendemic status poses a great risk of occupational exposure to HBV for HCWs and PSWs with regular blood–skin contact, although this risk has not been estimated in any Nigerian study. The risk of transmission from infected blood is said to be 100 times more for HBV than for HIV in non-immune individuals.[11] HBVc became part of the Nigerian National Programme on Immunisation for children aged 0–5 years in 2004.[12] Suboptimal immunisation coverage is however still a huge problem, especially in northern Nigeria.[12 13] To effectively control HBV in the Nigerian setting would therefore require a plurality of approaches. Prevention of new infections in at risk adults should complement prevention of perinatal transmissions. There is currently no universal HBVc programme for high risk adults in Nigeria. Such adults, however, can access HBVc individually in primary healthcare centres at subsidised rates.

The Federal Road Safety Corps (FRSC) was established by the Federal Government of Nigeria in 1988 due to the high rate of road traffic crashes in the country.[14] Road safety functions of FRSC include rescue and emergency care of road traffic crash victims, and this brings them into regular contact with blood.[14] All FRSC members participate in rescue operations, although this is more frequent for the marshal cadre. Crashed vehicles with broken glass increase the risk of sharps injuries for these PSWs. This exposes them and the accident victims they rescue to a high risk of HBV infection in this hyperendemic setting.

There are no studies on HBVc coverage of PSWs in Nigeria. The objectives of this study were to estimate HBVc coverage, and knowledge and sociodemographic determinants of full dose uptake in FRSC members, Kaduna State, Nigeria, to inform relevant targeted vaccination policies.

## METHODS
### Study design
A quantitative cross sectional survey of FRSC members, Kaduna Sector Command (KSC), Nigeria, was carried out.

### Setting and target population
Kaduna State is the third most populous state in Nigeria and has three senatorial zones with 23 local government areas.

The FRSC is divided into 12 zonal commands; each zonal command has sector commands which are subdivided into unit commands (UCs).[15] There are currently 204 UCs in Nigeria.[15] The first 11 UCs are located in the KSC, with the KSC headquarters making them 12; these cover the entire 23 local government areas in Kaduna State (table 1).

KSC is one of the four sector commands that make up the RS1 zonal command whose headquarters is in Kaduna. Two major cadres exist in FRSC: officer and marshal, although the latter is subdivided into marshal inspectorate and road marshal assistant. At the time of this study, there were 789 FRSC members in KSC, 26% of whom were officers and 74% marshals. The study was conducted in six randomly selected UCs: KSC headquarters, Saminaka, Kakau, Gwantu, Katari and Kachia.

### Inclusion and exclusion criteria
Only FRSC members in KSC aged ≥18 years with ≥6 months of service were included in the study. This ensured that only adults long enough in service to be made aware of the risk of HBV were surveyed. FRSC members in RS1 zonal command headquarters were excluded from the study as the zonal command includes other states beyond the study scope.

### Sample size
This was estimated using the formula for cross sectional surveys: $n = 1.96^2 \times p(1-p)/d^2$, where n is the required sample size, p is prevalence estimate of HBVc in previous studies and d is precision or acceptable error margin (5%).[16] Ogoina *et al*'s prevalence estimate of 36.2% in a survey of 290 HCWs in Nigeria[17] was used as a proxy

**Table 1** Location, coverage and staff distribution of unit commands of Federal Road Safety Corps, Kaduna Sector Command, Nigeria, June–July 2015

| Commands | Designation | Staff strength | | | No of LGAs covered | Location (LGA) |
| | | Cadre | | | | |
| | | Officer | Marshal | Total | | |
|---|---|---|---|---|---|---|
| Kaduna Sector Command Headquarters | RS1.1 | 46 | 118 | 164 | 2 | Kaduna North |
| Kafanchan UC | RS1.11 | 15 | 44 | 59 | 4 | Jama'a |
| Birnin Gwari UC | RS1.12 | 17 | 35 | 52 | 1 | Birnin Gwari |
| Zaria UC | RS1.13 | 24 | 66 | 90 | 5 | Sabon Gari |
| Saminaka UC | RS1.14 | 10 | 36 | 46 | 1 | Lere |
| Sabon Tasha UC | RS1.15 | 16 | 52 | 68 | 1 | Chikun |
| Kakau UC | RS1.16 | 18 | 66 | 84 | 2 | Chikun |
| Birnin Yero UC | RS1.17 | 15 | 44 | 59 | 1 | Igabi |
| Gwantu UC | RS1.18 | 8 | 31 | 39 | 2 | Sanga |
| Katari UC | RS1.19 | 19 | 37 | 56 | 1 | Kachia |
| Kachia UC | RS1.110 | 10 | 26 | 36 | 2 | Kachia |
| Tashan Yari UC | RS1.111 | 10 | 26 | 36 | 1 | Makarfi |
| | Total | 208 | 580 | 789 | 23 | |

LGA, local government areas; RS, Road Safety; UCs, unit commands.

as there is no existing study on PSWs in Nigeria. Anticipating a lower prevalence rate among non-HCWs with an expectedly lower level of awareness of HBVc, 30% prevalence was assumed ($n=1.96^2 \times 0.3(1-0.3)/0.5^2=323$). Using 24% as the anticipated non-response rate (q),[17] a final sample size of 425 was estimated using the formula: $N_f = N_s/1-q$, where $N_f$ is the final sample size and $N_s$ the initial sample size.[18]

## Sampling

The sampling frame was a list of the 12 UCs from the KSC headquarters. Each UC was considered a cluster. Clusters were randomly selected using a computer generated set of random numbers until the sample size was achieved. This simple random selection of clusters was to ensure representativeness of selected UCs.[19] Six UCs were selected for the study. All FRSC members in the selected UCs were targeted for questionnaire distribution.

## Data collection

UCs of FRSC have compulsory weekly meetings. Permission was obtained for data collection at these meetings. The UCs were informed of the research prior to visits. Data were collected between 17 June and 22 July 2015. The participant information sheet was reviewed with the staff, with emphasis on voluntary participation, anonymity and confidentiality of the collected data. Inclusion criteria and implied consent were further explained; completion of the questionnaire was considered consent to participate. Participants were asked to seal their completed questionnaires in the given envelopes and drop them in a common collection box provided by the researcher. This was to ensure anonymity. Those unwilling to participate

were asked to drop the sealed uncompleted questionnaires in the box alongside participants. Non-respondents were therefore not identified during data collection. Two UCs (KSC Headquarters and Kakau) were revisited in subsequent meetings due to poor initial attendance. Routine attendance lists taken by the UCs at the initial meetings were used to prevent re-participation of previous participants.

## Instruments

Due to the paucity of studies on the research topic, accessing a prevalidated questionnaire for the study was difficult. After an extensive literature search, only Al-Hussami's 'Hepatitis B vaccine knowledge and acceptance' questionnaire could be found.[20] This questionnaire has been used for HCWs in the USA. It was validated in two pilot studies with testing for inter-reliability but the test statistic was not reported.[20] There were 44 multiple choice questions, including some open ended ones. A structured anonymous self-administered questionnaire was adapted from this questionnaire for the present study (see online supplementary appendix A). Only questions relevant to the research questions were selected. Questions were simplified to suit the literacy status of the study population. The adapted study questionnaire contained 17 questions that elicited information on demographics (sex, age, duration of service, cadre and rank), HBV knowledge and perception of risk of exposure, and HBVc knowledge and status. Although rank was obtained, this was not included in the analysis as it mirrors cadre. The questionnaire was pilot tested on FRSC members in RS1 zonal command headquarters.

## Statistical analysis

Table 2 describes the variables in the study.

All analyses were conducted using SPSS V.21. Descriptive analysis ascertained frequencies and distributions of data. Histograms showed both HBV knowledge and HBVc knowledge scores to be normally distributed, hence their mean and SD were calculated as was the percentage of participants scoring above the mean scores. As the outcome variable (HBVc status) was binary, logistic regression analysis was used in testing for associations with the independent variables (table 2).[21] To mitigate confounding, univariate analyses were first carried out, and the variables identified as significantly associated (p<0.05) with HBVc status were included in the multivariate analysis for independent predictors of full dose HBVc.[21] Adjusted odds ratios (AOR) with 95% CI for each variable were computed and the significance level was set at p<0.05. Missing data on each variable were excluded in the analysis of the variable.

## RESULTS

There were 354 questionnaires distributed in the six UCs sampled from FRSC, KSC. Six questionnaires were discarded for having missing data on up to three of the independent variables or on the dependent variable and ≥2 independent variables. Seven questionnaires were submitted blank. In total, 341 completed questionnaires were included for analysis, giving a response rate of 96.3%. Online supplementary appendix C shows the percentage of missing data for each of the 14 questions analysed. Missing data were most frequent on the question on recommended dose of vaccine (9.7%; 33/341) followed by the question on the duration of protection from full dose HBVc (8.5%; 29/341). All participants provided data on cadre. Most respondents were men. aged 30–39 years, had worked between 3 and 10 years with FRSC and were of marshal cadre (table 3).

## HBV knowledge

The mean total number of correct answers to the HBV knowledge questions was 3.0 out of 6.0 (SD 1.5). Only 46% (157/341) of participants scored above the mean. The proportion of correct answers to HBV knowledge questions ranged from 2.1% (7/337) on route of transmission of HBV to 93.2% (317/340) on having ever heard of HBV. Approximately 22.6% (76/337) of respondents answered 'I don't know' to the question pertaining to the route of transmission of HBV, and this response was the most frequent. Only 2.1% (7/337) correctly identified contact with infected blood and blood contaminated body fluid as routes of transmission of HBV. Only 4 participants (1.2%; n=341) answered all six HBV knowledge questions correctly while 16 (4.7%, n=341) answered none correctly. HBV infection was perceived as more serious than HIV by most respondents (56.7%; 190/335) and about 3.0% (10/335) felt it was less serious than HIV. While 20.6% (69/335) claimed no knowledge of the

seriousness of HBV compared with HIV, 19.7% (66/335) ascribed equal severity to the two.

## HBVc knowledge

The mean number of correct answers to HBVc questions was 2.0 out of 4.0 (SD 1.1). Approximately 42.2% (144/341) of participants had scores higher than the mean score. All four questions on HBVc were answered correctly by only 4.1% (14/341) of participants while no correct answer was given by 11.7% (40/341). Rate of correctness ranged from 6.1% (19/312) for the question on duration of protection from full dose HBVc, to 86.6% (291/336) for the question on having ever heard of HBVc. Most respondents (62.9%; 210/334) described HBVc as very effective; 6.9% (23/334) rated it slightly effective, 2.7% (9/334) felt it was not effective at all and 27.5% (92/334) indicated not knowing its effectiveness. Approximately 54.9% (169/308) of respondents correctly identified recommended full HBVc dose as ≥3 doses while 1.6% (5/308) and 3.9% (12/308) thought it was 1 dose and 2 doses, respectively. Up to 39.6% (122/308) indicated not knowing the recommended full dose of HBVc.

## Perception of risk of occupational exposure to HBV

While most respondents (55.3%; 188/340) rated themselves at high risk of occupational exposure to HBV, 22.4% (76/340) did not know their risk status. Whereas 5.3% (18/340) of respondents considered themselves at no risk of exposure to HBV, 5.9% (20/340) and 11.2% (38/340) rated themselves at low and moderate risks of exposure, respectively. After dichotomising this variable into 'no risk perceived' and 'risk perceived' categories, 72.4% (246/340) had some level of risk perception while 27.6% (94/340) had no risk perception for HBV.

## HBVc coverage

Of the 341 participants, 6 did not provide data on their HBVc status. Ten others answered 'yes' to having ever received HBVc but omitted the number of doses received and were therefore inputted as missing data. Only 325 respondents (95.3%) were included in the subanalysis. Approximately 60.9% (198/325) of the respondents affirmed having ever received ≥1 dose of HBVc and 50.0% of these (99/198) had received ≥3 doses, resulting in full dose coverage of 30.5% (99/325) among the respondents. Approximately 39.1% (127/325) of respondents had never received HBVc. Together with the 99 participants with <3 doses, 69.5% (226/325) were classified as 'not vaccinated' while 30.5% (99/325) were labelled 'vaccinated'.

## Logistic regression analyses

All of the variables were significantly associated with HBVc on univariate analyses (table 4) and were included in the multivariate analysis for independent predictors of full dose HBVc uptake (table 5). Being a woman was associated with about twice the likelihood of having received full dose HBVc (table 5). When risk perception was analysed as a dichotomous variable ('no risk perceived' vs

**Table 2** Description of variables in the study, Federal Road Safety Corps, Kaduna Sector Command, Nigeria, June–July, 2015

| Variable | Description | Type of data |
|---|---|---|
| **Independent variables** | | |
| Sex | This was the sex of the study participants, categorised as either men or women | Nominal |
| Age | This variable ascertained the age of participants on their last birthday. It was categorised to enhance anonymity from 18 years, which is the age definition of commencement of adulthood, to ≥50 years, which marks the age before retirement from Nigerian Civil Service at 60 years. The categories included: 18–29 years; 30–39 years; 40–49 years; ≥50 years | Ordinal |
| Duration of service | This variable elicited how long a respondent had been in service with the Federal Road Safety Corps. It was categorised into: 6 months–2 years (probation period in the civil service); 3–10 years; 11–19 years; and ≥20 years (close to retirement by service year at 35 years). | Ordinal |
| Cadre | This ascertained the official class of participant based on position and seniority in office. There were two major categories: officers and marshals, with the latter sub-categorised into marshal inspectorate and field marshal assistant in descending order. It also signified educational qualification order with the least educated being the field marshal assistant. | Nominal/Ordinal |
| Risk perception | This ascertained the level of perception of occupational risk of exposure to hepatitis B virus by respondents. It was initially categorised as: no risk of exposure, low risk of exposure, moderate risk of exposure, high risk of exposure and I don't know. This was later dichotomised for further analysis by merging the 'I don't know' group with the 'no risk' group to form a 'no risk perceived' category with the rest forming the 'risk perceived' category. | Nominal/Ordinal |
| Hepatitis B virus knowledge score | This variable sought to estimate the level of knowledge of basic information on hepatitis B virus. It includes questions on hepatitis B virus awareness, seriousness compared with HIV and route of transmission. For each participant, the number of questions answered correctly was noted as the score (see scoring table in online supplementary appendix B). | Continuous |
| Hepatitis B vaccination knowledge score | This measured the level of basic knowledge of hepatitis B vaccination among participants. It comprised questions on hepatitis B vaccination awareness, effectiveness, recommended full dosage and duration of protection from full dose vaccination. For each participant, the number of questions answered correctly was noted as the score (see online supplementary appendix B). | Continuous |
| **Dependent variable** | | |
| Hepatitis B vaccination status | Information was elicited on whether the participant had ever received hepatitis B vaccination and the number of doses received. Descriptive analysis was done using these data. Dichotomisation of data was also done for logistic regression analysis. As only those with ≥3 doses of hepatitis B vaccination uptake are considered fully protected,[11] those with ≥3 doses were labelled 'vaccinated' and the rest 'not vaccinated'. This was noted as the hepatitis B vaccination status of each participant. | Nominal |

**Table 3** Sociodemographic characteristics of the study sample of Federal Road Safety Corps members, Kaduna Sector Command, Nigeria, June–July 2015

| Variable | Frequency | Percentage |
| --- | --- | --- |
| Sex (n=327) | | |
| Men | 260 | 79.5 |
| Women | 67 | 20.5 |
| Age (years) (n=338) | | |
| 18–29 | 64 | 18.9 |
| 30–39 | 167 | 49.4 |
| 40–49 | 87 | 25.7 |
| ≥50 | 20 | 5.9 |
| Duration of service (n=339) | | |
| 6 months–2 years | 36 | 10.6 |
| 3–10 years | 188 | 55.5 |
| 11–19 years | 87 | 25.7 |
| ≥20 years | 28 | 8.3 |
| Cadre (n=341) | | |
| Officer | 96 | 28.2 |
| Marshal | 245 | 71.8 |
| -Marshal inspectorate | 111 | 32.6 |
| - Field marshal assistant | 134 | 39.3 |

'risk perceived'), those with any level of risk perception for occupational exposure to HBV were about three times more likely to have received full dose HBVc than those without risk perception for HBV (table 5). Although the odds of being fully vaccinated increased with duration of service, this was not statistically significant. While HBV knowledge was not a significant predictor of full dose HBVc, knowledge of HBVc was significantly associated with full dose HBVc, with each unit increase in number of correct answers being associated with up to three times increased likelihood of being fully vaccinated (table 5).

In summary, full dose HBVc was 30.5% while ≥1 dose coverage was 60.9%. Female sex, perceiving their occupation as conveying a risk of HBV and increasing HBVc knowledge were significant independent predictors of full dose HBVc uptake among members of FRSC in Kaduna State, Nigeria.

## DISCUSSION

Like all other studies on HCWs in Nigeria,[22–25] this study has further demonstrated a high HBVc initiation rate (60.9%) with low completion rate (30.5%). This completion rate implies that only 30.5% of members of the FRSC, KSC, were adequately protected against HBV infection.[11] This means that almost 70% of these rescue workers perform their duties without adequate protection from HBV. This also places the accident victims whom they seek to rescue (including children) at risk of infection with HBV from infected FRSC members. This could

lead to an unbroken cycle of infectivity, morbidity and mortality from HBV in a nation still struggling with the HBV scourge. Poor uptake of HBVc among those at occupational risk of exposure to HBV in Nigeria is a common observation across studies.[22–28] Adoption of a universal HBVc policy in the country for HCWs and vulnerable PSWs could improve vaccine uptake. FRSC members are exposed to blood and sharp injuries from accident scenes. In a nation with a high prevalence of chronic HBV,[5] HBVc coverage of 30.5% for FRSC members, KSC, is low. A seroprevalence study to investigate the actual prevalence of HBV in these PSWs for appropriate intervention is therefore recommended.

Among FRSC members, women were 2.28 times more likely to be fully vaccinated against HBV compared with their male counterparts. Osazuwa-Peters *et al* observed a similar but insignificant female preponderance in HBVc among dental professionals in Edo State, Nigeria.[29] In contrast, Adekanle *et al*, in their survey of HCWs in Ile-Ife, Nigeria, observed 1.8 times increased chances of men receiving HBVc compared with women,[27] although this finding may be due to the men in the study being mainly doctors who had the advantage of professional knowledge of HBVc. In the present study, women may have been exposed to HBVc knowledge during antenatal hospital visits and while taking their children for immunisation.

More than 50% of respondents perceived themselves at high risk of occupational exposure to HBV. Disturbingly though, close to a quarter of participants claimed ignorance of their risk status. Together with those who perceived themselves to be at no risk, 27.6% of FRSC members had no risk perception for occupational exposure to HBV while 72.4% perceived themselves at risk. This falls within the range of 30–78% risk perception observed in studies among HCWs in Nigeria.[22 28] Despite the high risk perception rate, approximately 28% with no risk perception for HBV is quite disconcerting from a public health perspective considering the blood–skin exposure that rescue of accident victims could entail. Those who perceived themselves at risk of occupational exposure to HBV were three times more likely to be vaccinated than those without risk perception. All risk categories had higher odds of vaccination compared with those without risk perception. It is therefore important that FRSC members understand the risk of exposure to HBV (even if they feel it is low) as this appears to increase their likelihood of getting vaccinated.

Knowledge of HBV and HBVc among study participants was poor. Less than 47% of participants scored above the mean knowledge scores for HBV and HBVc. Knowledge was poorest for the route of transmission of HBV and duration of protection from full dose HBVc. Not knowing the route of transmission of HBV means that FRSC members might take inadequate precautions against HBV during rescue operations. It could also lead to stigmatisation of FRSC members already infected with HBV due to the wrong assumption of infectivity through

**Table 4** Univariate binary logistic regression analyses showing unadjusted OR of the association between independent variables and full dose hepatitis B vaccination uptake among Federal Road Safety Corps members, Kaduna Sector Command, Nigeria, June–July 2015

| Independent variable | Sample size | OR | 95% CI (p value) |
|---|---|---|---|
| Sex | 313 | | |
| Men | | 1 | |
| Women | | 2.66 | 1.51 to 4.70 (0.001) |
| Age (years) | 323 | | |
| 18–29 | | 1 | |
| 30–39 | | 2.08 | 0.97 to 4.44 (0.059) |
| 40–49 | | 3.30 | 1.47 to 7.40 (0.004) |
| ≥50 | | 5.67 | 1.84 to 17.50 (0.003) |
| Duration of service | 323 | | |
| 6 months–2 years | | 1 | |
| 3–10 years | | 5.69 | 1.31 to 24.72 (0.020) |
| 11–19 years | | 9.48 | 2.12 to 42.35 (0.003) |
| ≥20 years | | 13.39 | 2.61 to 68.56 (0.002) |
| Cadre | 325 | | |
| Officer | | 1 | |
| Marshal inspectorate | | 1.69 | 0.92 to 3.09 (0.091) |
| Road marshal assistant | | 2.10 | 1.18 to 3.74 (0.012) |
| Risk perception for occupational exposure to HBV | 324 | | |
| I don't know | | 1 | |
| No risk | | 3.92 | 0.78 to 19.63 (0.096) |
| Low risk | | 11.33 | 2.94 to 43.63 (<0.001) |
| Moderate risk | | 9.61 | 2.85 to 32.43 (<0.001) |
| High risk | | 11.07 | 3.87 to 31.70 (<0.001) |
| Risk perception for occupational exposure to HBV | 324 | | |
| No risk perceived | | 1 | |
| Risk perceived | | 7.39 | 3.27 to 16.71 (<0.001) |
| HBV knowledge score | 325 | 1.37 | 1.15 to 1.62 (<0.001) |
| HBVc knowledge score | 325 | 2.97 | 2.16 to 4.08 (<0.001) |

HBV, hepatitis B virus; HBVc, hepatitis B vaccination.

casual contact with sweat and saliva. HBV knowledge score was however not an independent predictor of HBVc in the study. This conflicts with Adekanle et al's observation of twice increased likelihood of complete HBVc among those with good knowledge of HBV in their survey of HCWs in Ile Ife, Nigeria.[27] However, their study did not elicit information on HBVc knowledge, a potential confounder in the demonstrated association.

Despite 62.9% of respondents describing HBVc as very effective, only 6.1% knew that full dose HBVc gives protection for ≥20 years. Knowing that receiving ≥3 doses of the vaccine can provide lifetime protection from HBV could provide the incentive for full dose uptake. HBVc knowledge was the most significant and precise independent predictor of full dose HBVc in this study. This contradicts Ogoina et al's finding of no significant association between HBVc knowledge and full dose vaccination among HCWs

in two tertiary hospitals in Nigeria.[17] However, they did not ascertain knowledge of vaccine effectiveness and duration of protection from full dose vaccination.

Age, cadre and duration of service were not significantly associated with HBVc in this study. Izegbu et al observed more likelihood of HBVc with decreasing age,[30] while Sofola et al associated increasing cadre with HBVc.[31] In another instance, longer duration of service was demonstrated to be associated with HBVc.[32] All of these studies were among health professionals who were expected to have professional exposure to HBV and HBVc knowledge, unlike the present study population.

Educational programmes towards improvement in HBV and HBVc knowledge, and risk perception among FRSC members, is a recognised relevant public health intervention from this study. The programme can be included in the schedules of the already existing compulsory weekly

**Table 5** Multivariate binary logistic regression analysis for independent predictors of full dose hepatitis B vaccination uptake among members of Federal Road Safety Corps, Kaduna Sector Command, Nigeria, June–July, 2015

| Independent variable | Adjusted OR (n=309) | 95% CI (p Value) |
|---|---|---|
| Sex | | |
| Men | 1 | |
| Women | 2.28 | 1.15 to 4.52 (0.019) |
| Age (years) | | |
| 18–29 | 1 | |
| 30–39 | 1.40 | 0.47 to 4.18 (0.542) |
| 40–49 | 0.99 | 0.28 to 3.55 (0.987) |
| ≥50 | 1.08 | 0.20 to 5.76 (0.931) |
| Duration of service | | |
| 6 months–2 years | 1 | |
| 3–10 years | 2.12 | 0.39 to 11.41 (0.384) |
| 11–19 years | 2.73 | 0.45 to 16.59 (0.276) |
| ≥20 years | 5.25 | 0.68 to 40.47 (0.112) |
| Cadre | | |
| Officer | 1 | |
| Marshal inspectorate | 1.60 | 0.77 to 3.33 (0.208) |
| Road marshal assistant | 0.87 | 0.41 to 1.85 (0.720) |
| Risk perception for occupational exposure to HBV | | |
| I don't know | 1 | |
| No risk | 2.93 | 0.47 to 18.41 (0.251) |
| Low risk | 7.12 | 1.47 to 34.47 (0.015) |
| Moderate risk | 4.50 | 1.03 to 19.63 (0.045) |
| High risk | 3.90 | 1.08 to 14.09 (0.038) |
| Risk perception for occupational exposure to HBV | | |
| No risk perceived | 1 | |
| Risk perceived | 2.86 | 1.06 to 7.70 (<0.001) |
| HBV knowledge score | 1.03 | 0.80 to 1.31 (0.843) |
| HBVc knowledge score | 2.68 | 1.83 to 3.92 (<0.001) |

HBV, hepatitis B virus; HBVc, hepatitis B vaccination.

inhouse training/manpower development of staff and in the routine basic training programme for new staff. Such enlightenment would be an inexpensive and easy intervention to improve HBVc uptake. Existing evidence on the positive impact of educational intervention on vaccine uptake is however weak.[33] The educational intervention should therefore be rigorously evaluated to ascertain its impact on HBVc uptake in FRSC members.

### Study strengths and limitations

This was a descriptive cross sectional survey which limits its suitability for demonstrating temporal relationships between explanatory and outcome variables.[34] It nevertheless showed independent associations useful in understanding predictors of full dose HBVc in this study population which will inform relevant public health interventions. Recall bias is another limitation of this retrospective study design as participants may not have remembered accurately their vaccination history, thereby introducing information bias.[35] This was mitigated at the analysis stage by omitting inconsistent data suggestive of guessing. Simple random sampling using the staff register would have yielded a more representative sample[21]; disproportionate distribution of sociodemographic variables such as sex and cadre in the study population made this infeasible.

The response rate of 96.3% was however impressive and minimises selection bias, thereby enhancing the generalisability of the research findings by improving the external and internal validity of the study.[36] The presence of Unit Commanders and other senior members of staff at the meetings during data collection and their participation in the research could have contributed to the high response rate. Subordinates who ordinarily may have declined participation could have felt a psychological obligation

to participate with their bosses. This power influence was minimised by the use of the participant information sheet which emphasised voluntary participation, and by anonymous data collection procedures. While anonymity and self-administration of the questionnaire could lessen social desirability bias, the use of social desirability scale would have been more appropriate in demonstrating this bias for appropriate statistical control.[37] Researcher bias was minimised by the use of a pre-validated questionnaire and by predetermining analytical strategies before data collection.[21] Possible exchange of information among participants could have introduced information bias. This was mitigated by the presence of the researcher during data collection with prior emphasis on non-communication between participants. Some participants who claim ignorance of HBV might have a different designation for the disease in the local language. This could bias the findings on HBV knowledge.

Although the study's questionnaire was adapted and pilot tested to reflect the study context, it was not tested for inter-rater reliability and validity within the study population. Missing data reduced the sample size for the multivariate analysis from 341 to 309. This was less than the pre-study estimate (323) and could lack sufficient power to detect significant associations, hence predisposing to type II error.[21] However, it constitutes a randomised 39% (309/789) of the study population, which is a good representation.[38] Confounding, a known menace in observational studies, was minimised at the analytical stage through multivariate logistic regression.[21] The research estimates on the association of varied levels of risk perception with full dose HBVc had very wide 95% CIs. This could be due to random errors in the sample.[36] A larger sample size in future studies could yield more precise estimates.

## Conclusion

Controlling HBV transmission is an important public health issue internationally, and in Nigeria, where the virus is hyperendemic. HBV infection is a preventable disease, and prevention is best achieved with HBVc.[6] FRSC members come into regular contact with blood and are at risk of contracting HBV. HBVc coverage among FRSC members in Kaduna State, Nigeria is low (30.5%). Knowledge of HBV and HBVc is poor in this study population. Female sex, perceiving there to be an occupational risk of exposure to HBV and increasing HBVc knowledge are independent predictors of HBVc uptake among FRSC members, KSC. Educational intervention aimed at improving awareness of the occupational risk of HBV and the importance of HBVc is required to improve HBVc coverage among this vulnerable group of PSWs. Recommended future studies include: a qualitative study to ascertain FRSC members' perception of HBVc and subjective reasons for non-uptake of the vaccine; a seroprevalence study to determine the actual immune status of FRSC members in KSC and estimate the prevalence of HBV in this study group for appropriate intervention; and validation of the questionnaire in the Nigerian context with pre-testing and re-testing for reliability.

**Acknowledgements** The University of Liverpool is acknowledged for providing the platform for this research. We thank the Commonwealth Scholarship Commission for paying the tuition fee for the MPH study of the corresponding author, thereby enabling this dissertation research. We appreciate the staff and management of the Federal Road Safety Corps, Kaduna Sector Command, for their cooperation and participation in this study.

**Contributors** This study was carried out as a dissertation research by CLO under the close supervision of CMB, in partial fulfilment of the requirement for the award of the degree of Master of Public Health by the University of Liverpool, UK. CLO collected the data and conducted the analyses. These were reviewed by CMB. The manuscript was drafted by CLO and reviewed and revised by CMB. Both authors approved the final version for publication.

**Funding** This research received no specific grant from any funding agency in the public, commercial or not-for-profit sectors.

**Competing interests** None declared.

**Patient consent** The study was a cross sectional survey using anonymous self-administered questionnaires. Implied consent was used wherein completion of the questionnaire was regarded as consent to participate. This was clearly written in the participant information sheet and was also verbally explained to participants by the researcher prior to data collection. Participation was voluntary. No participant identifying information was included in the questionnaire. Age and duration of service were categorised to enhance anonymity. Data were collected from six Unit Commands (UCs) of the Federal Road Safety Corps, Kaduna Sector Command. Names of UCs were not included in the questionnaire and data were inputted together into the SPSS. This was to further enhance anonymity. The dataset is therefore totally anonymous.

**Ethics approval** This study was approved by the University of Liverpool's Ethics Committee and the Ethics Committee of Kaduna State Ministry of Health.

**Provenance and peer review** Not commissioned; externally peer reviewed.

**Data sharing statement** Extra data can be accessed via the Dryad data repository at http://datadryad.org/ with the doi:10.5061/dryad.545q0.

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
