## [Reviewer comments · BMJ Open]

ARTICLE DETAILS

TITLE (PROVISIONAL)	Hepatitis B Vaccination Coverage, Knowledge and Socio-Demographic Determinants of Uptake in High-Risk Public Safety Workers in Kaduna State, Nigeria: A Cross-Sectional Survey
AUTHORS	Ochu, Chinwe; Beynon, Caryl

VERSION 1 - REVIEW

REVIEWER	Tamara Giles-Vernick Director of Research/Group Leader Institut Pasteur France
REVIEW RETURNED	09-Jan-2017

GENERAL COMMENTS	This manuscript provides an analysis of a cross-sectional study of hepatitis B vaccination coverage, knowledge, and socio-demographic determinants among workers for the Federal Road Safety Corps. The manuscript provides a clear justification and argument, employs a methodology (questionnaire) adapted to local circumstances, and targets a population of health care workers who have not yet been studied with regards to HBV knowledge and vaccination. The results are not especially surprising, in that HBV vaccination coverage and HBV knowledge remain poor among this population. The authors also acknowledge clearly the advantages and shortcomings of their study and seek to make concrete recommendations based its findings. The manuscript requires major revisions to make it publishable, but could potentially contribute to sub-Saharan African studies of HBV knowledge and vaccination. The manuscript is rather weak in its understanding of HBV epidemiology in sub-Saharan Africa, and the authors need to account for its specificities. I think that the problem results in part from the authors' reliance at critical moments in the manuscript on HBV studies from wealthy countries, where risk factors for infection with HBV are significantly different from those in sub-Saharan Africa (see, for instance, Lavanchy 2008; CDC 2011; Boal, no date
---

provided; Woodruff et al 1993). They contend that those with chronic HBV are at an elevated risk (15-25%) of premature mortality from HCC and cirrhosis, and that the “Prevention of new HBV infections in adulthood is a recognized global public health priority.” The authors seem to be unaware that age at first exposure and infection is crucial to this outcome – a factor that is especially important in sub-Saharan Africa, where HBV is highly endemic. There, the vast majority of transmission occurs during the first two years of life. By the time people have reached adulthood in sub-Saharan Africa, well over 70% have already been exposed to HBV. Moreover, children infected in early life run the greatest risk (90%) of becoming chronic carriers. It is these chronic carriers who are at elevated risk for developing hepatocellular carcinoma (HCC) and cirrhosis. In addition, the claim that preventing new adult infections of HBV *in Africa* is a priority does not square with my reading of this literature (see Lemoine, Eholié and Lacombe, 2015).

It is true, as the authors rightly note, that those who have never been exposed to HBV can be infected through sexual intercourse or through exposure to blood products. Nonetheless, the risk of these adults becoming chronic carriers is *very small* during adulthood. Hence, the risk of cirrhosis or hepatocellular cancer among these adults is very low. In short, the authors have not addressed a fundamental feature of HBV in Africa and should substantially revise the justifications for the study, some aspects of the analysis, and recommendations.

My specific recommendations are as follows:

1. The authors need to re-evaluate risk of infection among health care workers in light of this epidemiology.
2. The authors should rely more heavily on major African studies and sub-Saharan policy recommendations that have been published recently. Lemoine et al (2015) in the *Journal of Hepatology* offers some very useful insights into the problem of HBV and viral hepatitis in Africa in general, as well as policy recommendations. Stockdale et al (2015) in *Transactions of the Royal Society of Tropical Medicine*, as well as Schweitzer et al (2015) in the *Lancet* offer reliable and up-to-date estimates of African prevalence. Recent articles in *BMC Infectious Disease* (and *American Journal of Tropical Medicine and Hygiene* (Debes et al 2016) elsewhere provide comparative statistics of HCW prevalence of HBV.

I would recommend that categorical claims like “Perception of threat of a disease provides cue for action in favour of a health-promoting behavior” (citing a study examining hemodialysis patients, who are vastly different from Nigerian health workers!) need serious nuancing. A very rich medical anthropology literature shows that how people understand “threat”, “disease” and “health-promoting behaviors” can be highly variable, and the logics underpinning their quests for health are not always self-evident. It would be important to

keep in mind that such models are always not universally transposable, and that applying insights from hemodialysis patients with access to complex care to a population of Nigerian health workers ought to raise questions.

3. The authors need to re-think the recommendations that they make in light of the particularities of HBV in sub-Saharan Africa. Specifically, vaccinating *all* FRSC workers may not make sense, if over roughly 70% have already been exposed in childhood. Authors would thus need to consider the cost-effectiveness of the recommendation for vaccinating all health care workers in the FRSC. Given that most adults to be vaccinated in the FRSC would have already been exposed to HBV as children (probably >70%), would it be more cost-effective to vaccinate all members, or to screen them to determine exposure and then to vaccinate only those who have not yet been exposed?
Some analysts might argue that in a low-resource country like Nigeria, redoubling efforts for the routine vaccination of young children would be a better use of limited resources. Others might argue that screening and diagnosis and linkage to care of adult chronic carriers would be more cost-effective (for a study of a screen-and-treat program, see Lemoine et al, *Lancet Global Health*, 2016).
4. A few lines about the recent history of vaccination and its availability and accessibility would be important to include. Nigeria integrated the HBV vaccination into its Expanded Programme on Immunisation in 2004. But when did the vaccine against HBV become available to the rest of the population? Is it now widely available? Do all health structures/vaccination centers have sufficient vaccine stocks? What are the associated costs with the three-dose vaccine?
5. A little more detail about what public safety workers do would be helpful to readers.
6. In the section on study strengths and limitations, the authors carefully acknowledge the limitations of their study and seek to correct for it. The authors do try to correct for faulty recall, but they don't acknowledge the possibility that at times, people respond in ways that they think they ought to respond. This may not be screened out through the omission of inconsistent responses. In addition, that HBV knowledge is low among these workers is not surprising. While I recognize that the authors cannot re-conduct the study, it might be useful to mention that HBV could be known and described as another diagnostic entity in a local language. This could potentially affect findings about HBV knowledge.
7. The authors should include something more about the ethical procedures followed in the study. Did participants receive an information notice and sign an informed consent form? (Authors should explain what "implied consent" meant in practice).

REVIEWER	Tekalign Deressa University of Gondar, Ethiopia
REVIEW RETURNED	24-Jan-2017

GENERAL COMMENTS	Reviewer report Ochu et al have done an interesting research of Survey of Hepatitis B Vaccination Coverage, and Knowledge and Socio-Demographic Determinants of vaccine Uptake. This study could benefits if the authors consider the points indicated below. Minor revision  1. What is the designation RS for? 2. It would be more informative if the authors rewrite the result section of the abstract in a simpler language and by including some demographic information. I would prefer to see “Any dose hepatitis B vaccination coverage was 60.9%; full-dose coverage was 30.5%” presented in a different way. Same is true for the second sentence. 3. The statement “Institutionalizing free hepatitis B vaccination could improve uptake 26 among FRSC members” on Page 4, Line 25 and 26 (conclusion section) is out of context as the authors have no data on whether lack of access to HBV vaccine could contribute to the low vaccine up take. In fact, because it is expensive is presented as an alternative answer for the question on the reason for not taking HBV vaccine. However, I haven’t seen the responses that implicate the conclusion they have drawn. 4. Page 5, L52: This?????....please revisit! 5. Try to avoid numerals at the beginning of a sentence. E.g. 354 participants were.... Instead write three hundred 6. I am not quite sure how significant the questions like “How serious do you think being infected with hepatitis B virus is compared to HIV? “ Seriousness in terms of what ?
---

	Pathogenicity, virulence, transmission rate? 7. Some of the findings especially those that contradict the previous finding need to be discussed. 8. Do all FRSC in all cadre level have equal risk of exposure to the blood because of their profession? Overall, the authors put together very interesting data that may be of significance for those who have closely related interest. However, the authors need to extensively revise their manuscript before publication.
--	--

VERSION 1 – AUTHOR RESPONSE

Reviewer 1: Tamara Giles-Vernick

1. The introduction section has been revised to provide a clear communication of the justification for the study based on the epidemiology of hepatitis B virus (HBV) infection in sub-Saharan Africa (SSA). Though chronicity is the major HBV outcome of interest in hyperendemic regions, prevention of new infections in high-risk adults is a necessary complementary strategy in achieving effective control of the virus. This is more so considering the growing mortality trend of acute viral hepatitis. Lozano et al. demonstrated about 11 times higher rise in age-standardized global mortality rate for acute viral hepatitis than for hepatocellular carcinoma (HCC) between 1990 and 2010 while noting a decline in percentage mortality for hepatic cirrhosis.[1] Acute viral hepatitis is the main cause of the rare but deadly fulminant hepatic failure in SSA.[2] Furthermore, infected healthcare workers (HCWs) and public safety workers (PSWs) can readily infect children in the course of their duties and these in turn could become chronic carriers. Control of HBV in a hyperendemic setting therefore demands plurality of approaches targeting not just children but high-risk adults.

2. The authors initially cited studies on public safety workers (not healthcare workers) in developed countries because there are no studies, to the best of our knowledge, on this study population in SSA. Studies by Boal et al. (the year 2005 was indicated in the initial submission) and Woodruff et al. (1993) are not cited in the revised manuscript. Rischitelli et al.'s (2001) systematic review of the risk of contracting hepatitis B or C among PSWs is retained as well as Lavanchy's (2008) discussion of chronic viral hepatitis as a global public health issue. The authors have however laid more emphasis on studies in SSA. We found the recommended articles by Lemoine et al. (2015), Schweitzer et al. (2015), and Stockdale et al. (2015) very useful and have included these, along with some other recent studies in SSA, in the revisions. The claim that "Perception of threat of a disease provides cue for action in favour of a health-promoting behaviour" has been deleted along with the cited Adams et al.'s article on hemodialysis patients as these are not relevant in the present study.

3. The authors have deleted the recommendation for the Federal Government of Nigeria to provide free hepatitis B vaccination (HBVc) for all unvaccinated Federal Road Safety Corps (FRSC) members and to enact a policy to institutionalize free mandatory vaccination for newly recruited staff as these recommendations are not supported by the research findings. Given the low HBVc coverage recorded

in the study, the authors instead recommend a sero-prevalence study to investigate the actual prevalence of HBV among FRSC members for appropriate intervention. This is contained in page 27, lines 317-319 of the revised manuscript.

4. The reviewer rightly noted that HBVc became integrated into the National Programme on Immunization in Nigeria in the year 2004. This covers only children 0-5 years. There is no universal coverage for at-risk adults. These adults however can access HBVc in any primary healthcare centre at subsidized rates. This information is contained in pages 7-8, lines 106-113 of the revised manuscript. Some tertiary hospitals (like the Ahmadu Bello University Teaching Hospital, Zaria where the Corresponding Author works) provide these vaccinations for as low as 100NGN per dose to these at-risk adults. Information on whether participants were aware of these provisions was not elicited in this study and could be beneficial in future studies.

5. Examples of public safety workers have been included in page 6, lines 75-76 of the revised manuscript.

6. Social desirability bias has been discussed and presented in page 31, lines 406-408. Information bias could also result from participants claiming ignorance of HBV because they have a different designation for the disease in their local language. This is discussed in page 31, lines 413-414 of the revised manuscript.

7. A letter of permission was gotten from the Zonal Commanding Officer RS1 Zonal Command to collect data at the compulsory weekly meetings of FRSC members in Kaduna Sector Command. The Zonal Command informed the Unit Commands in Kaduna Sector Command of the research permit. Prior to visiting a Unit Command, the Corresponding Author (CA) made two telephone calls to the Unit Commander first to confirm the date and then as a reminder. This was to ensure availability of staff for participation in the research. FRSC members had prior information of the data collection date. On the recruitment date, the CA was allowed to address the staff before data collection. The participant information sheet (PIS) was reviewed with the staff. The purpose of the study was explained with emphasis laid on voluntary participation, anonymity and confidentiality of collected data. Inclusion criteria and implied consent were well explained to participants. Completion of questionnaire was considered consent to participate. This was clearly written in the PIS and on the first page of the questionnaire, and was further emphasized verbally by the CA. The questionnaire had no participant-identifying information; it also had no provision for designation of Unit Command. Age and duration of service were categorized to further enhance anonymity. Participants were instructed to seal completed questionnaires in the provided envelopes and drop them in a common collection box provided by the CA. This was to ensure anonymity. Those not willing to participate were asked to drop the sealed uncompleted questionnaires in the box alongside participants. Non-respondents were therefore not identified during the data collection process. This information is summarized in pages 12-13, lines 175-185 of the revised manuscript.

Reviewer 2: Tekalign Deressa

1. RS stands for Road Safety. The acronym has been explained in lines 11 (abstract) and 141 (foot note on table 1, page 9) of the revised manuscript.

2. Revisions have been made to the result section of the abstract vis: "Any dose hepatitis B vaccination coverage was 60.9%; full-dose coverage was 30.5%. Less than 47% of participants scored above hepatitis B virus (HBV) and hepatitis B vaccination mean knowledge scores" are replaced with "Most participants were males, aged 30-39 years, with 3-10 years of service, and of Marshal cadre. HBVc coverage was 60.9% for ≥ 1 dose and 30.5% for ≥ 3 doses. Less than 47% of participants scored above the mean knowledge score for hepatitis B virus (HBV) and HBVc." This is

seen in page 3, lines 17-20 of the revised manuscript.

3. The statement “Institutionalizing free hepatitis B vaccination could improve uptake among FRSC members” has been deleted from the conclusion section of the abstract as there are no data to support this claim from the present study.

4. The statement “This results in >2 million deaths from chronic liver diseases annually” has been deleted from the introduction section of the main text due to its ambiguity and irrelevance as a justification for the study. Though the statement was meant to attribute more than 2 million global deaths annually to liver cirrhosis and hepatocellular carcinoma, this may not be relevant for new adult HBV infections where the risk of chronicity is minimal. This revision is contained in page 5, line 55 of the revised manuscript.

5. Revisions have been made as recommended; sentences no longer start with numerals throughout the manuscript.

6. The question “How serious do you think being infected with hepatitis B virus is compared to HIV?” was meant to test the lay understanding of participants of the infectiousness and likely outcome of being infected with either of the two viruses in the Nigerian setting. The adapted questionnaire was simplified to suit the lay and literacy status of the study population who were not healthcare workers and were mostly of Marshal cadre (having low literacy status). The term ‘serious’ was used in its colloquial sense based on the Nigerian context. HBV is 100 times more infectious than HIV [3]. Using the term ‘infectiousness’ or ‘infectious’ could have confused some of the participants. In Nigeria, being infected with HBV could be considered more ‘serious’ than being infected with HIV. Access to HIV testing and treatment is free in Nigeria while there is no such provision for HBV. Patients infected with HBV pay out-of-pocket for their treatment. Since screening is not free, most people do not realise they are chronic carriers of HBV until they present with chronic liver diseases. Persons with chronic HBV are probably more likely to die from complications than those with HIV in the Nigerian context. Pilot study showed good understanding of the question by all participants. This was noted during the debriefing session of the pilot study. However, it is not unlikely that this subjective knowledge question could have biased the HBV knowledge assessment of FRSC members. Outcome of acute infection with HBV most times could be more favourable than for HIV. Validation of the questionnaire in the Nigerian context with pretesting and retesting for reliability is recommended for future studies.

7. Summary of negative findings have been included in page 30, lines 372-377 of the revised manuscript.

8. All cadres of FRSC members are involved in rescue operations at road traffic crash scenes. However, the more senior Officers do more office work than the Marshals. Hence, the Marshals could be assumed to be more occupationally exposed to HBV than the Officers. This assumption though demands more scrutiny in future studies for justification.

References:

1. Lozano R, Naghavi M, Foerhan K, et al. Global and Regional Mortality from 235 Causes of Death for 20 Age Groups in 1990 and 2010: A Systematic Analysis for the Global Burden of Disease Study 2010. *Lancet* 2012;380:2095-128.
2. Jayakumar S, Karvellas C. Fulminant Viral Hepatitis. *Critical Care* 2013;29:677-97.
3. Centers for Disease Control and Prevention (CDC). Immunization of Health-Care Personnel:

Recommendations of the Advisory Committee on Immunization Practices (ACIP). Morbidity and Mortality Weekly Report (MMWR): Recommendations and Reports 2011;60:1-45.